

# Time-dependent variational Monte Carlo study of the dynamic response of bosons in an optical lattice

**Mathias Gartner[1][⋆], Ferran Mazzanti[2] and Robert E. Zillich[1]**

**1** Institute for Theoretical Physics, Johannes Kepler University Linz,
Altenberger Straße 69, 4040 Linz, Austria
**2** Departament de Física i Enginyeria Nuclear, Campus Nord B4-B5,
Universitat Politècnica de Catalunya, E-08034 Barcelona, Spain

⋆ mathias.gartner@jku.at

## Abstract

We study the dynamics of a one-dimensional Bose gas at unit filling in both shallow and deep optical lattices and obtain the dynamic structure factor $S(k, \omega)$ by monitoring the linear response to a weak probe pulse. We introduce a new procedure, based on the time-dependent variational Monte Carlo method (tVMC), which allows to evolve the system in real time, using as a variational model a Jastrow-Feenberg wave function that includes pair correlations. Comparison with exact diagonalization results of $S(k, \omega)$ obtained on a lattice in the Bose-Hubbard limit shows good agreement of the dispersion relation for sufficiently deep optical lattices, while for shallow lattices we observe the influence of higher Bloch bands. We also investigate non-linear response to strong pulses. From the power spectrum of the density fluctuations we obtain the excitation spectrum, albeit broadened, by higher harmonic generation after a strong pulse with a single low wave number. As a remarkable feature of our simulations we furthermore demonstrate that the full excitation spectrum can be retrieved from the power spectrum of the density fluctuations due to the stochastic noise inherent in any Monte Carlo method, without applying an actual perturbation.


# 1   Introduction

The dynamic structure factor $S(k, \omega)$ is a fundamental quantity as it contains the maximal information about the dynamics of many-body quantum systems that one can obtain by inelastic scattering [1], such as the excitation energies $\omega(k)$ and the lifetime of collective excitations. In quantum gases $S(k, \omega)$ can be measured by Bragg spectroscopy [2], with relative momentum and energy resolution similar to inelastic neutron scattering in condensed matter [3]. The calculation of $S(k, \omega)$ is a demanding task beyond very simple Hamiltonians or approximations, such as the Bijl-Feynman model [4, 5], or the Bogoliubov-de Gennes technique in the mean field limit [6, 7]. Advanced variational methods based on action minimization, such as the correlated basis function approach [8] or the multi-configuration time-dependent Hartree algorithm [9], can achieve much more accurate results. All these methods are reliable in many cases but are not expected to work well in all situations. Monte Carlo methods, on the other hand, are known to be able to produce statistically exact predictions for bosons, although this only applies to the ground state at zero temperature [10], or to static ensemble averages at finite temperature [11]. Access to the excitation spectrum is restricted to the evaluation of the dynamic response in imaginary time and its reconstruction in frequency space by inverting the Laplace transform. This is a rather difficult procedure as Laplace inversion is a well known ill-posed mathematical problem, worsened in practice by the fact that the stochastic noise of the simulation is exponentially amplified in the result. The way to tackle these problems is to build many reconstructions of the response and to use stochastic methods based on simulated annealing [12] or genetic algorithms [13, 14] to produce an approximate dynamic structure factor. While this method can yield good results, it is computationally very expensive and usually gets only the broad features, not resolving well the fine details of the response. Other methods available for dynamic simulations are either restricted to lattice systems, like time-evolving block decimation [15], nonequilibrium dynamical mean-field theory [16], and the time-dependent density matrix renormalization group method [17–19], or they work best in one dimension, like methods based on continuous matrix product states [20]. Consequently, accurate methods that allow for time dependent simulations of strongly correlated many-body systems which can describe the linear, but also nonlinear response to perturbations, are in demand.

The time-dependent variational Monte Carlo (tVMC) method [21–24] is particularly suitable for the study of quantum many-body dynamics, allowing for perturbations of any strength. It can be applied to analyze many different situations, such as ramping up the lattice depth [25] or interaction quenches [22], as well as many-body dynamics far from equilibrium [26]. It has also been extended to wave functions based on artificial neural networks [27, 28]. In this work we use tVMC to analyze the dynamic response of a Bose gas to a probe pulse in an optical lattice in one dimension, where we use a continuous space representation rather than the Bose-Hubbard limit. We present a new way to calculate the dynamic structure factor $S(k, \omega)$ of strongly interacting bosons in continuous space, based on tVMC simulations of the time evolution after weak pulses. For strong pulses, we enter the nonlinear regime. A strong perturbation with only a single wave number creates excitations with multiplies of this wave number due to higher harmonic generation. We exploit this to obtain an approximation of the full excitation

spectrum by analysing the power spectrum of the density fluctuations after such a strong pulse with a single low wave number. Finally, we introduce a third way to calculate the excitations with tVMC: surprisingly, we can obtain the excitation spectrum from the power spectrum of the density fluctuations with no perturbation at all, i.e. from the tVMC time evolution of the variational ground state, thanks to the stochastic noise inherent in Monte Carlo simulations.

## 2 Method

We use tVMC to study the response of the Bose gas, initially in the ground state at time $t = 0$, to an external perturbation $\delta V_p(x, t)$, and monitor the time evolution of the density fluctuations $\delta\rho(x, t) = \rho(x, t) - \rho(x, 0)$. In the linear response regime, the ratio of their Fourier transforms is the density response function, with its imaginary part being the dynamic structure factor [29]. We perform a series of simulations for a system of $N$ identical bosons of mass $m$, moving in a one dimensional optical lattice $V(x)$ and interacting via a contact potential. The Hamiltonian reads

$$H = \sum_{i=1}^{N} \left( -\frac{\hbar^2}{2m} \frac{\partial^2}{\partial x_i^2} + V(x_i) + \delta V_p(x_i, t) \right) + g \sum_{i<j}^{N} \delta(x_i - x_j), \tag{1}$$

with the coupling constant $g$ parametrizing the strength of the two-body interaction. As usual for cold atomic systems, where the optical lattice potential is generated by counter propagating laser beams with wave number $k_L$, we will use the form $V(x) = V_0 \sin^2(k_L x)$ for the potential [30], which corresponds to a lattice constant of $\pi/k_L$. Throughout this work we will report $V_0$ and $g$ in units of the recoil energy $E_r = \hbar^2 k_L^2/2m$ and $E_r/k_L$, respectively, and we use $x_0 = \pi/k_L$ and $t_0 = \hbar/E_r$ as length and time unit.

In the deep lattice limit where the amplitude $V_0$ is large, $H$ can be approximated by the lattice Hamiltonian of the single-band Bose-Hubbard model (BHM) [31, 32]

$$H_{\text{BHM}} = -J \sum_{i<j}^{N} b_i^\dagger b_j + U/2 \sum_i^N n_i(n_i - 1), \tag{2}$$

where $b_i^\dagger$, $b_i$ and $n_i$ are the creation, annihilation and number operator for bosons at lattice site $i$. For given $V_0$ and $g$ in Eq. (1), the on-site interaction $U$ and the hopping parameter $J$ of the BHM can be evaluated numerically performing band-structure calculations [33]. Within our continuous space tVMC simulations, we can access both the BHM regime and the region of shallow optical lattices, where the single-band BHM is no longer valid.

### 2.1 Model wavefunction

The tVMC method relies on a model wave function with variational parameters that are propagated in time. For modeling the time-dependent wavefunction $\Phi(\boldsymbol{x}, t)$ of the many-body system, with $\boldsymbol{x} = (x_1, \dots, x_N)$, we use a Jastrow-Feenberg ansatz [34] with one- and two-particle correlation functions

$$\Phi(\boldsymbol{x}, t) = e^{\sum_i^N u_1(x_i, t)} e^{\sum_{i,j}^N u_2(x_i - x_j, t)}. \tag{3}$$

In tVMC simulations, we parametrize the wavefunction by a set of time-dependent complex variational parameters $\boldsymbol{\alpha}(t) = \{\alpha_1(t), \alpha_2(t), \dots, \alpha_P(t)\}$ and it is convenient to write the wavefunction in the form

$$\Phi(\boldsymbol{x}, \boldsymbol{\alpha}(t)) = \exp\left( \sum_K \mathcal{O}_K(\boldsymbol{x}) \alpha_K(t) \right), \tag{4}$$

where every variational parameter $\alpha_K(t)$ is coupled to a local operator $\mathcal{O}_K(\boldsymbol{x})$ [21]. For these local operators we use third order B-splines [35], which are piecewise polynomial functions, restricted locally to intervals $Y_{mp}$ centered at the points of a uniform grid. We use two sets of intervals, the first ($m = 1$) on a uniform grid in $[0, L]$ for the one-body function $u_1$, and the second set ($m = 2$) on a grid in $[0, L/2]$ for the two-body correlations $u_2$, where $L$ is the size of the simulation box. For each interval $Y_{mp}$ we denote the corresponding spline by $B_{mp}(x)$ and define the corresponding sets of operators $\mathcal{O}_{1p}(\boldsymbol{x}) = \sum_i^N B_{1p}(x_i)$ and $\mathcal{O}_{2p}(\boldsymbol{x}) = \sum_{i<j}^N B_{2p}(|x_i - x_j|)$. Using this form of the local operators in equation (4), together with the index mapping $K \equiv (m, p)$ we get

$$\Phi(\boldsymbol{x}, \boldsymbol{\alpha}(t)) = \exp\left(\sum_p^{P_1} \sum_i^N B_{1p}(x_i)\alpha_{1p}(t)\right) \exp\left(\sum_p^{P_2} \sum_{i<j}^N B_{2p}(|x_i - x_j|)\alpha_{2p}(t)\right). \qquad (5)$$

By exchanging the summation in the exponentials we can identify the one- and two-particle correlation functions of the Jastrow-Feenberg ansatz (3) as $u_1(x_i, t) = \sum_p^{P_1} B_{1p}(x_i)\alpha_{1p}(t)$ and $u_2(x_i - x_j, t) = \sum_p^{P_2} B_{2p}(|x_i - x_j|)\alpha_{2p}(t)$, respectively.

The effect of the contact interaction in the Hamiltonian (1) has been directly incorporated in the wavefunction by using an appropriate boundary condition on $u_2$ for $x_i = x_j$, according to [36]. In particular, we impose a condition on the variational parameters such that the logarithmic derivative of the wavefunction satisfies $\frac{1}{\Phi}\frac{\partial}{\partial x_i}\Phi = \frac{1}{4}k_L g$ for any $x_i = x_j$, which originates from the solution of the two-body problem with contact interaction in one dimension.

As shown in [21], the equations governing the time evolution of the variational parameters are

$$\mathrm{i}\sum_{K'} S_{KK'}\dot{\alpha}_{K'} = \langle \mathcal{E}\mathcal{O}_K \rangle - \langle \mathcal{E} \rangle \langle \mathcal{O}_K \rangle , \qquad (6)$$

with the correlation matrix $S_{KK'} = \langle \mathcal{O}_K \mathcal{O}_{K'} \rangle - \langle \mathcal{O}_K \rangle \langle \mathcal{O}_{K'} \rangle$ and the local energy $\mathcal{E} = \frac{H|\Phi\rangle}{|\Phi\rangle}$. These coupled nonlinear ordinary differential equations can be solved numerically, where in every time step the expectation values forming the coefficient matrix $S_{KK'}$ and the right hand side of the equation system are calculated by Monte Carlo integration. In all the simulations presented in this work we use 400 ($P_1 = P_2 = 200$) complex variational parameters $\alpha_K$, which we have checked to be enough to produce converged results.

## 2.2 Monte Carlo sampling and time propagation

In order to accomplish a stable time propagation we need to reduce the numerical errors that are built up during the time evolution of the system. To achieve this, we pre-condition and regularize the matrix $S_{KK'}$ before solving the Eqs. (6). As a first step we scale the matrix by $S'_{KK'} = S_{KK'}/\sqrt{S_{KK}S_{K'K'}}$ and as a second step we add a small regularizing factor $\varepsilon$ to the diagonal entries ($S'_{KK'} \rightarrow S'_{KK'} + \varepsilon\delta_{KK'}$) in order to prevent instabilities due to eigenvalues that are close to zero in $S'_{KK'}$ [37]. The same value $\varepsilon = 10^{-4}$ was used in all simulations. To solve the resulting system of equations we use a QR decomposition and a fourth order Runge-Kutta scheme to propagate the differential equations (6) in time. We found that a stable time propagation requires a reasonably small time step of at least $\delta t = 10^{-4} t_0$, which we used throughout this work. The Monte Carlo estimates for $S_{KK'}, \langle \mathcal{E}\mathcal{O}_K \rangle, \langle \mathcal{E} \rangle$ and $\langle \mathcal{O}_K \rangle$ are obtained using the Metropolis-Hastings algorithm, and a total of $N_{\mathrm{MC}} = 12500$ uncorrelated samples are used in every time step of the numerical propagation of Eq. (6). The density observable $\langle \rho(r, t) \rangle$, which is the main quantity of interest in our simulations, is calculated at every hundredth simulation time step, i.e. at multiples of a time step $\delta t_\rho = 0.01 t_0$. In order to get the density estimate with high accuracy we use $N_{\mathrm{MC},\rho} = 2.5 \cdot 10^6$ uncorrelated samples.

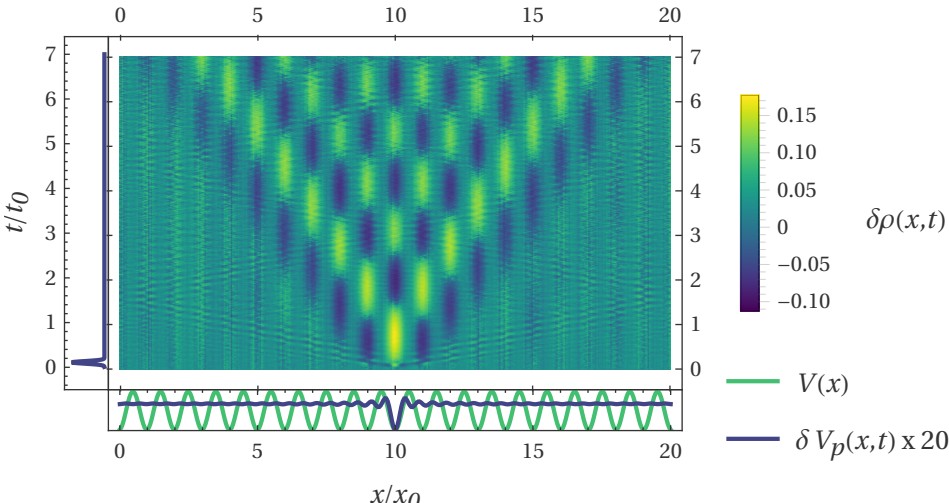

Figure 1: Spatial density fluctuation $\delta\rho(x,t) = \rho(x,t) - \rho(x,0)$ evolving in time, from which we obtain $S(k,\omega)$. Initially the system is in the ground state (obtained via i-tVMC) of the optical potential $V(x)$ (green line). At time $t = 0$, a weak perturbation pulse $\delta V_p(x,t)$ with a Gaussian time profile (blue line along vertical axis) and a superposition of various momentum modes (blue line along horizontal axis), given by equation (8), is turned on. The main color map shows the propagation in time (vertical) and space (horizontal axis) of the density fluctuation $\delta\rho$ induced by the perturbation. The amplitude of the pulse $\delta V_p$ is magnified by a factor of 20. The system parameters $V_0 = 7\,E_r$, $g = 0.41\,E_r/k_L$ (corresponding to $U/J = 6$) and the pulse parameters $V_e = 0.0125\,E_r$, $t_e = 0.1\,t_0$ and $\tau = 0.04\,t_0$ are used in this simulation.

## 3 Results

For the calculations we proceed as follows: we first perform tVMC simulations in imaginary time (i-tVMC) with $\delta V_p = 0$ to obtain the variational ground state of the Hamiltonian in Eq. (1). The result is then used as the initial wavefunction for the real time simulation, where we turn on the perturbing potential $\delta V_p$ at $t = 0$ and monitor the density fluctuations $\delta\rho(x,t)$ (see Fig. 1). If the perturbation is weak, we use linear response theory [29] to estimate the dynamic structure factor

$$S(k,\omega) = -\frac{1}{\pi}\,\text{Im}\left[\frac{\delta\tilde{\rho}(k,\omega)}{\delta\tilde{V}_p(k,\omega)}\right], \tag{7}$$

where $\delta\tilde{\rho}(k,\omega)$ and $\delta\tilde{V}_p(k,\omega)$ are the space and time Fourier transforms of the density fluctation and the perturbing potential, respectively.

For comparison with exact ground state results, we also performed i-tVMC calculations in the absence of the optical lattice ($V_0 = 0$), leading to the Lieb Liniger model [36]. The resulting ground state energy compares well to the energy obtained in Bethe ansatz calculations (see Ref. [38, Eq. (10)]), with a relative error of less than 0.4% for the range of interaction strengths $g$ used in this work.

### 3.1 Linear response

We calculate the dynamic structure factor $S(k,\omega)$ from Eq. (7) for several values of the coupling strength $g$ and the optical lattice amplitude $V_0$. We use $N = 20$ particles with a density $n = 1/x_0$ in a simulation box of size $L = x_0 N$ with periodic boundary conditions, corresponding to unit filling. To excite the system we apply a multi-mode pulse with a Gaussian time

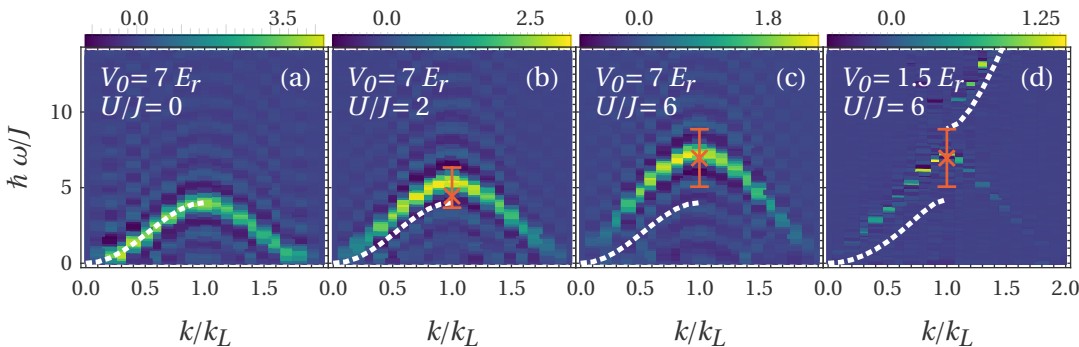

Figure 2: Dynamic structure factor $S(k,\omega)$ from tVMC simulations for bosons in an optical lattice of amplitude $V_0 = 7 E_r$ and equivalent BHM parameter $U/J = 0; 2; 6$ (panels (a)–(c)), and $V_0 = 1.5 E_r$ with $U/J = 6$ (panel (d)). The dashed white lines show the dispersion of non-interacting particles ($U = 0$) for the given value of $V_0$, obtained from band structure calculations. The red bar indicates the spread of the multipeak feature in $S(k,\omega)$ of Ref. [39].

profile

$$\delta V_p(x,t) = V_e\, e^{-(t-t_e)^2/\tau^2} \sum_{j}^{j_{\max}} \sin^2\left(k_j x\right), \tag{8}$$

where the spatial part is a superposition of up to $j_{\max} = 40$ modes with wave numbers given by $k_j = 2\pi j/L$. In particular, we choose $V_e = 0.0125 E_r$, $t_e = 0.1 t_0$ and $\tau = 0.04 t_0$. This pulse imparts an energy less than 0.25% of the ground state energy to the system, which shows that the perturbation is weak enough for linear response theory to apply. To check this further we doubled $V_e$ and indeed got the same $S(k,\omega)$. In the linear regime we can get the full excitation spectrum in a single tVMC simulation since modes are excited simultaneously, but independently of each other. The short pulse length $\tau$ also ensures that it excites a broad range $2\pi/\tau$ of energies. In any case, the pulse in Eq. (8) can be easily tailored, to excite only selected modes if required.

We present in Fig. 2 the dynamic structure factor $S(k,\omega)$, in units of $k_L$ and $J/\hbar$ for $k$ and $\omega$, respectively. Panels (a)–(c) show $S(k,\omega)$ for a deep optical potential $V_0 = 7 E_r$ and interaction strengths $g = 0; 0.14; 0.41 E_r/k_L$, corresponding to the ratios $U/J = 0; 2; 6$ of the BHM, respectively. Panel (d) shows $S(k,\omega)$ for a shallow lattice with $V_0 = 1.5 E_r$ and $g = 2.8 E_r/k_L$, corresponding to the equivalent BHM ratio $U/J = 6$. White dashed lines denote the Bloch dispersion of non-interacting particles. The tVMC result for $U/J = 0$ in panel (a) demonstrates that the peaks in $S(k,\omega)$ reproduce the exact non-interacting Bloch dispersion perfectly. The broadening of the tVMC dispersion, as well as the ringing oscillations, are artifacts resulting from the Fourier transform over a finite simulation time of length $T = 10 t_0$. When we increased $T$ and thus the computational cost, the artificial oscillation frequency increased and the amplitude decreased. As $U/J$ is increased, the excitation energies increase also, and the dispersion becomes linear for small $k$. The positions of the peaks in $S(k,\omega)$ as function of $\omega$ are in good agreement with results of [39] obtained by exact diagonalization of the BHM. The details of our $S(k,\omega)$, however, differs from the results in [39], where multiple close peaks were obtained for $N = 16$. In panels (b) to (d), the spread of these peaks is indicated by a red bar, with the central main peak of [39] indicated by a cross. The main difference of our system compared to [39] is that we use continuous coordinates instead of using the Hubbard approximation leading to the discrete lattice of the single-band BHM. Furthermore we simulate a slightly higher number of particles and use a variational description of the wavefunction.

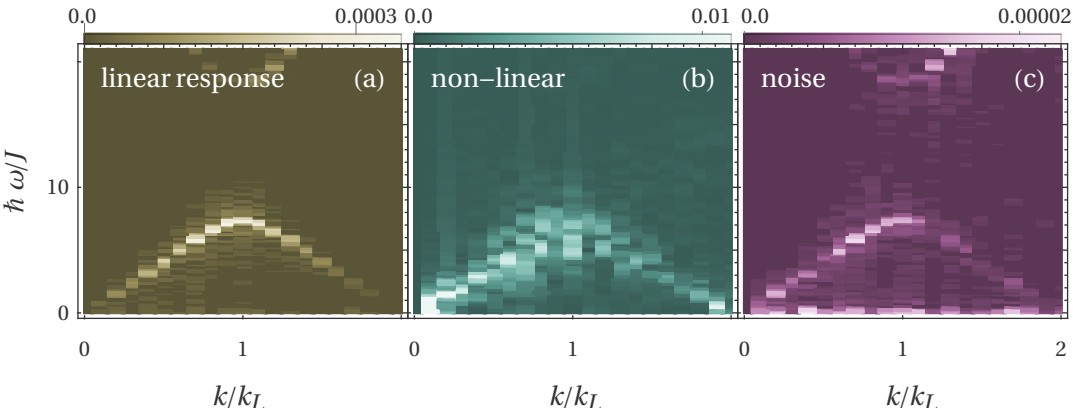

Figure 3: Square root of the power spectrum $|\delta\tilde{\rho}(k,\omega)|$ of the density fluctuations obtained with three different tVMC simulation variants. Panel (a) shows the response to a weak multi-mode pulse of the form given in Eq. (8). In panel (b), the system is excited with a single mode at $k = 0.1\,k_L$ with an amplitude as large as the lattice potential ($V_e = V_0$). In panel (c), no perturbation is applied and the density fluctuations in the time propagation are solely due to the stochastic noise in the Monte Carlo simulation. In all three simulations the lattice amplitude is $V_0 = 3\,E_r$ and the BHM parameter is $U/J = 6$.

For the shallow optical lattice case shown in panel (d), corresponding to $V_0 = 1.5\,E_r$, the band gap is comparable with the band width. In such a case the single-band BHM does not apply. The dispersion is linear over a wider range of $k$ values than in the deep lattice, while the maximum of the first band hardly changes. In such a shallow lattice, we also observe an energy increase of the second band with respect to the Bloch dispersion. Notice that the seemingly smaller broadening of the curve in panel (d) is due to the fact that all energies are expressed in units of $J$, which is $J = 0.04\,E_r$ for $V_0 = 7\,E_r$ and $J = 0.16\,E_r$ for $V_0 = 1.5\,E_r$.

## 3.2 Nonlinear response

The tVMC method is not restricted to weak perturbations, and thus one can use it to explore the response of the system outside the linear regime. In order to demonstrate this, we again perturb the same Bose system at unit filling in the optical lattice with $V_0 = 3\,E_r$ and $U/J = 6$, but this time with a strong pulse. Instead of exciting all wave numbers simultaneously with the weak pulse in Eq. (8), we excite only the lowest mode compatible with the periodic boundary condition, with wave number $k_1 = 2\pi/L = 0.1\,k_L$, but with a pulse strength equal to the amplitude of the lattice potential, $V_e = V_0$. We use a pulse length $\tau$ five time longer than in the linear response simulations previously described, and also set $t_e = 0.5\,t_0$ to move the peak of the pulse to larger times for a smooth switch-on of the perturbation. Overall, the integrated pulse strength is 30 times stronger than that of the weak multi-mode pulse to compare with. Outside the linear regime, $S(k,\omega)$ no longer describes the full response of the system to the perturbation, and furthermore, for $k \neq k_1$ we have $\delta\tilde{V}_p(k,\omega) = 0$, and thus $S(k,\omega)$ cannot even be calculated. Therefore we show the square root of the power spectrum, $|\delta\tilde{\rho}(k,\omega)|$.

The panel (b) of Fig. 3 shows $|\delta\tilde{\rho}(k,\omega)|$ after the strong pulse with wave number $k_1$. As it could be expected, a very pronounced peak in the non-linear response appears at $k_1$. However, the strong pulse excites a wide range of multiples of $k_1$ via higher harmonic generation. For comparison, in panel (a) of Fig. 3 we show $|\delta\tilde{\rho}(k,\omega)|$ for the weak multi-mode pulse $\delta V_p(x,t)$ of Eq. (8). Note that in the linear response regime $|\delta\tilde{\rho}(k,\omega)|$ conveys the same information as $S(k,\omega)$, see the previous Fig. 2. Compared to the linear response to the

weak multi-mode pulse, the non-linear response exhibits a much broader excitation band, but it essentially follows the dispersion relation obtained from linear response; the broadening is expected for higher harmonic generation in a system with a non-linear dispersion. Panel (b) of Fig. 3 demonstrates that a sufficiently strong long wavelength perturbation yields the full excitation spectrum, albeit with significant broadening.

### 3.3  Excitations from noise

An even more remarkable feature of the tVMC method is that the full excitation spectrum can also be obtained in the opposite limit, i.e. applying no perturbation at all. We can simply propagate the variational ground state in real time. The stochastic noise in $S_{KK'}$ and the right hand side of Eq. (6) produces fluctuations around the exact time evolution which we can use to calculate the excitation spectrum of all modes. Similarly to the nonlinear case, $S(k, \omega)$ is not accessible because $\delta \tilde{V}_p(k, \omega) = 0$, this time for all $k$. We show $|\delta \tilde{\rho}(k, \omega)|$, generated entirely by the stochastic noise, in panel (c) of Fig. 3. The peak locations giving the excitation energies are essentially identical to the linear response results shown in panel (a). In this way, the Monte Carlo noise can be effectively used to explore the excitation spectrum of the system, although as seen from the color scales in Fig. 3, the noise generated power spectrum is much weaker.

As expected, the noise is reduced when we increase the sample size per time step, but the signal-to-noise ratio of the density fluctuation power spectrum remains unchanged. If, on the other hand, we improve the variational ansatz $\Phi$, the parameter optimization with i-tVMC leads to a variational ground state closer to the exact ground state. When we increased the number of parameters $\alpha_K$, the signal-to-noise ratio in $|\delta \tilde{\rho}(k, \omega)|$ dropped, because improving the variational wave function reduces the variance of the local energy $\mathcal{E}$. The sampling noise in the quantities on the right hand side of Eq. (6) falls, while the sampling noise in $S_{KK'}$ on the left hand side is barely affected. In this way, there are less noise-induced perturbations to the ground state evolution of the parameters $\alpha_K$ when we solve Eq. (6). In the limit that the optimized ansatz $\Phi$ is the *exact* ground state, $\mathcal{E}$ is the exact ground state energy, with zero variance, while the correlation matrix $S_{KK'}$ is still non-zero and invertible, which leads to $\dot{\alpha}_K = 0$, thus there is no noise-induced time evolution. The only noise left in the power spectrum of the density fluctuation is the sampling noise which carries no information on the dynamics because it is uncorrelated between time steps. But apart from a few selected problems, the exact many-body wave function is not known, and in general there will always be some noise-induced time evolution of $\alpha_K$ about their optimized values.

Our result in panel (c) of Fig. 3 suggests a new simulation strategy where the unperturbed ground state is propagated in time, rather than exciting specific modes with suitable temporally and spatially shaped weak or strong external pulses. With this new type of simulation we can for example determine the excitation spectrum in a large range of $\omega$- and $k$-values, which is useful when analyzing a new system with little knowledge about the relevant range of energies and momenta to explore. It has the added benefit of not having to choose any specific form for the perturbation potential. We stress, however, that $|\delta \tilde{\rho}(k, \omega)|$ is not proportional to $S(k, \omega)$. We can obtain the excitation energies from the peaks of either of them, but the stochastic noise is not white and thus has different strength for different energies and momenta. For example, in the present case, the peaks in panel (c) of Fig. 3 are clearly smaller for $k > k_L$ than for $k < k_L$, while the linear response result in panel (a) looks more symmetric about $k_L$. In order to obtain the spectral weights of the dynamic structure function, we have to use linear response theory as demonstrated in section 3.1.

# 4 Conclusion

In summary, we have explored the possibility of using time-dependent variational Monte Carlo (tVMC) to obtain the dynamic structure factor $S(k, \omega)$, or more generally the excitation spectrum, of many-body quantum systems under the action of a pulsed perturbation. Specifically, we have analyzed the linear and nonlinear dynamics of a one-dimensional system of bosons in an optical lattice described by a continuous Hamiltonian. In both deep and shallow lattices, we explore several interaction strengths corresponding to the same ratio $U/J$ of the Hubbard interaction and hopping parameters, to assess the universality of the dependence of the excitation spectrum on it. For shallow lattices and as expected, we observe a deviation from the single-band Bose-Hubbard result, with the dispersion being linear over a wider range of momenta. However, for the lowest band, the excitation energy at the edge of the Brillouin zone is remarkably universal.

Besides the weak perturbation regime where linear response theory applies, we have also explored the dynamics after a very strong perturbation, and the dynamics with no perturbation at all. In the latter case we simply propagate the optimized ground state in real time to obtain the excitation spectrum from the fluctuations due to the stochastic noise intrinsic to every Monte Carlo method. This can be useful when studying complex systems where the nature of the excitations is not known, and the right choice of the perturbation operators is not so obvious. In order to explore the non-linear regime, we apply pulses coupling to a single mode, but with peak strengths of the order of the optical lattice depth itself. These strong pulses excite the full range of wave numbers via higher harmonic generation. This could be relevant for Bragg spectroscopy of the excitation spectrum, since only one or few momentum transfers need to be chosen to obtain an approximate $S(k, \omega)$ for a wide range of momenta.

## Acknowledgments

M. G. and R. E. Z thank G. Carleo and M. Holzmann for fruitful discussions and acknowledge computational resources of the Scientific Computing Administration at Johannes Kepler University.

**Funding information** F. M. acknowledges financial support by grant PID2020-113565GB-C21 funded by MCIN/AEI/10.13039/501100011033, and from Secretaria d'Universitats i Recerca del Departament d'Empresa i Coneixement de la Generalitat de Catalunya, co-funded by the European Union Regional Development Fund within the ERDF Operational Program of Catalunya (project QuantumCat, ref. 001-P-001644).

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
