# Peer review of "Time-dependent variational Monte Carlo study of the dynamic response of bosons in an optical lattice"

_SciPost Physics, doi:SciPost Phys. 13, 025 (2022)_

## Round 1 · Referee Report · Anonymous (Referee 1) · 2022-3-30

Report

In the submitted manuscript, Gartner et al. discuss the applicability of the time-dependent variational Monte Carlo (tVMC) method to calculate the dynamic structure factor S(k,w) of strongly interacting bosons in continuous space in both linear and non-linear regime, and in absence of perturbation.
The paper is well written but the discussion of the results is a bit brief. However, I may give a positive recommendation for a properly revised manuscript.

Requested changes

1) In the abstract, introduction and conclusions, authors claim to compute the dynamic structure factor S(k,w) with the tVMC method. However, when they discuss the non-linear and in absence of perturbation cases, they compute and show the space and time Fourier transform of the density $\delta\tilde{\rho}(k,w)$ only. I feel this is misleading. They should be clearer in both the introduction and the abstract.

2) In Fig.2 a color legend should be added in order to allow the reader to compare the results with Ref.[35]. Authors should explain while, as claimed in the manuscript, "the shape of S(k,w), however, differs somewhat from the results in [35], where a more complex structure was obtained for N=16". They also claim that "the broadening of the tVMC dispersion, as well as the ringing oscillations, are artifacts resulting from the Fourier transform over a finite simulation time window of length T=10t_0". Could they increase the simulation time in order to remove such effects or there are compulational limitation?

3) As the authors are presenting a new way to calculate the dynamic structure factor, I feel like they do not make clear the potential of their approach, in particular compared to exact diagonalization method (Ref.[35]), in terms of accuracy, limit on the size of the system and computation time.

4) Authors claim that "methods that allow for time dependent simulations of strongly correlated many-body systems which can describe the linear, but also nonlinear response to perturbation, are in demand". I agree with them, for this reason I feel like the more interesting part, the non linear regime, is too rushed and should be extended in order to make the manuscript more appealing.

  • validity: good
  • significance: good
  • originality: high
  • clarity: high
  • formatting: perfect
  • grammar: excellent

Author:  Mathias Gartner  on 2022-06-03  [id 2553]

(in reply to Report 1 on 2022-03-30)
Category:
answer to question

We thank the referee for reading the manuscript and his/her report. In the following we address the questions and comments raised:

1) We agree with the referee that this was indeed a bit confusing. The dynamic structure factor $S(k,\omega)$ is the ratio of the density fluctuation $\delta \tilde \rho(k, \omega)$ to the perturbation $\delta \tilde V_p(k, \omega)$ causing it, in the linear regime. In the case of the noise-generated signal, the Monte Carlo noise itself, which is not given as external perturbation, is the source of density fluctuations. Therefore, for comparison with the results from weak and strong pulses, we choose to present $|\delta \tilde \rho(k, \omega)|$, i.e. the square root of the power spectrum of the density fluctuations (which we call that way in the revised manuscript). For the strong pulses, we can still calculate $\operatorname{Im} \left[ \delta\tilde\rho(k, \omega) / \delta \tilde V_p(k, \omega) \right]$, but it would be misleading as the system would no longer be in the linear regime where different modes contribute independently. For example a strong pulse generates higher harmonics of $k$, without actually perturbing with higher $k$. In order to clarify this point, in the introduction, we now introduce the 3 cases (linear, nonlinear, no perturbation) and explain the choice of $|\delta \tilde \rho(k, \omega)|$ or $S(k,\omega)$, and also changed the abstract.

2) We added a color legend in Fig.2, however, a direct comparison with Ref.[35] (now [39]) will still not be possible due to the lack of such a color legend therein. The differences in $S(k,\omega)$ can be attributed to the fact that we use a variational method, a slightly different number of particles and a continuous Hamiltonian (in contrast to the BHM model of [35], now [39]). We add a more extensive discussion of this in the revised version. Regarding the second question,we have chosen a total simulation time of $T=10 t_0$ because for this time window we get enough energy resolution to see the relevant features present in the dynamic structure factor. There is no a priori technical restriction on increasing the simulation time. In order to ensure energy conservation for very long simulation times, a small propagation time step $\delta t$ is required, which additionally increases the computational workload. Increasing the total time $T$ reduces the strength of the artificial oscillations caused by the cut-off at $T=10 t_0$, but the cost associated with preserving the accuracy of the calculation would increase considerably. We added a sentence mentioning this explicitly.

3) The tVMC method is both variational and a Monte Carlo technique. Being variational, the accuracy of the prediction is ultimately limited by the accuracy of the wave function employed, which is determined by its functional form and by the actual number of parameters in the model. The accuracy of exact diagonalization methods are not limited by a model, but they are limited by the number of basis vectors employed, which increases exponentially with the number of degrees of freedom. That drastically limits the dimensionality, type of Hamiltonian, and number of particles that can be treated reliably, e.g. a lattice Hamiltonian and $N=16$ particles in case of Ref.[35], now [39]. Monte Carlo methods do not have these severe limitations, even less if they are variational as in tVMC. Monte Carlo methods are known to scale well with both the number of particles and the dimensionality of space, and are efficient for both discrete and continuous Hamiltonians. Our implementation of the tVMC method is readily portable to 2D and 3D geometries. We are currently working on that.

4) We agree with the referee that nonlinear dynamics is a very interesting point that deserves a deeper treatment. The number of possibilities for nonlinear perturbation as well as the aspects to study is large. We have already performed and are currently performing further simulations of the nonlinear response of bosons in optical lattices, among other systems. The technical challenges are similar to the linear case in that we need to make sure that all numerical parameters are under control (sample size and sample equilibration, time step, number of parameters for our Jastrow ansatz, simulation length), however the interpretation and assessment of the results is harder because nonequilibrium many-body dynamics is much less explored than linear response. The preliminary results are promising and look very interesting, but have to stand up against the mentioned numerical checks, which requires plenty of computer time, and also have to be critically assessed regarding the limitations of the variational ansatz. This is why it is prudent to put more work into the nonlinear studies and eventually devote a separate publication to nonlinear many-body dynamics in optical lattices. In the present paper, we therefore present only one nonlinear result, where we have afforded all the numerical checks.

---

## Round 1 · Referee Report · Anonymous (Referee 2) · 2022-4-6

Strengths

  1. Nice if somewhat limited extension of Carleo's time-dependent variational Monte Carlo work on bosons in one dimension to allow calculations of the dynamic structure factor.

  2. Interesting comparison of the effects of applying very small and very large time-dependent perturbations, exploring the linear and highly non-linear regimes and showing how the linear response can be seen even in the non-linear regime.

  3. Interesting use of the stochastic noise inherent in Monte Carlo simulations to calculate the dynamic structure factor without applying a time-dependent perturbation.

Weaknesses

  1. Little new physics. Advances reported are primarily technical.

  2. Unnecessarily confusing definition of the wave vectors of the periodic applied potentials.

Report

This paper uses the time-dependent variational quantum Monte Carlo method, introduced (I believe?) by Giuseppe Carleo, to calculate the dynamic structure factor of a system of bosons moving in a one-dimensional sinusoidal potential and interacting via delta-function contact interactions. For the structure factor calculations, an additional time-dependent sinusoidal potential is applied. Although the model studied is idealized, it is not a bad description of bosonic atoms in a cigar-shaped trap aligned with a one-dimensional optical lattice. All results presented are for unit filling, in which there is one boson per potential well.

The model studied - the Lieb-Liniger model with a sinusoidal applied potential - can be solved exactly using the Bethe Ansatz when the applied potential is zero and reduces to the Bose-Hubbard model when the applied potential is very strong. It has been so well investigated that I doubt this paper tells us anything new about the physics it represents.

The methods used are similar to those used in Ref. [23]: G. Carleo, L. Cevolani, L. Sanchez-Palencia, and M. Holzmann, "Unitary Dynamics of Strongly Interacting Bose Gases with the Time-Dependent Variational Monte Carlo Method in Continuous Space", Phys. Rev. X 7, 031026 (2017). Carleo and co-workers also studied the Lieb-Liniger model, although they did not calculate the dynamic structure factor or apply a sinusoidal potential. The current work improves on Carleo's approach by using splines to represent the one- and two-particle correlation functions, $f_1$ and $f_2$, and by the introduction of new methods to calculate the dynamic structure factor. The simulation cell contains only 20 bosons, which is probably too few to obtain results representative of the large-system limit; Carleo looked at up to 100 bosons.

On the whole, although the number of new physics results reported in this manuscript is limited, the structure factor calculations are new and technically interesting, especially the comparison of results obtained in the linear response (small perturbation) and highly non-linear regimes. The paper is quite well written and I enjoyed reading it. I think it deserves publication without major changes.

Requested changes

  1. For reasons I don't understand, you choose to write the lattice potential as

    $$V_0 \sin^2(k_L x) = (V_0/2) (1 - \cos(2 k_L x)),$$
    so the wave vector of the applied potential is $2k_L$, not $k_L$ as readers will expect. This explains, for example, why the spectra in Fig. 2 are plotted over the domain $0 \leq k/k_L < 2$ instead of $0 \leq k/k_L < 1$. The same confusing use of $\sin^2$ functions appears in the definition of the time-dependent perturbation. This small issue of notation confused me repeatedly as I read the manuscript and made it significantly harder to follow than it should have been. Since the constant shift in the $1 - \cos(2 k_L x)$ factor is of no importance, why not make your readers' lives easier by writing the applied potential as $V_0 \cos(k_L x)$ or $V_0 \sin(k_L x)$ and redefining $V_0$ and $k_L$ accordingly? If you would rather make fewer changes, you could write the potential as $V_0 \sin^2 (\frac{k_L x}{2})$.

  2. On p4., you discuss the interesting observation that you can use the stochastic noise inherent in the Monte Carlo simulation to calculate the dynamic structure factor without applying a periodic perturbation. (Is this related to the fluctuation-dissipation theorem?) Have you thought about what would happen in the limit in which the trial wavefunction is exact, so that the local energy $\mathcal{E}(\mathbf{x})$ is a constant, equal to the ground-state energy regardless of the boson position vector $\mathbf{x}$. According to Eq. (2), the coefficients of the wavefunction then cease to evolve in time, which makes sense as you already have the exact ground state and are not applying a time-dependent perturbation. It also implies, however, that the fluctuations you are using to see the structure factor have disappeared. I suspect that your "no perturbation" method only works because your wave function is not very accurate, and that it will become harder and harder to use as your trial wavefunction improves. If that is correct, the no-perturbation approach is an interesting curiosity but may not be as useful as you hope. This issue may merit some discussion.

  3. On p2, the lead up to Eq.(3) says: "The correlation functions $f_1$ and $f_2$ depend on time-dependent complex variational parameters .... We can thus write (2) as ..." Strictly, I think Eq.(3) requires more than this: it relies on the fact that the logarithms of $f_1$ and $f_2$ depend linearly on the time-dependent variational parameters, so the current wording is misleading. Please amend.

  4. The expression for the local energy $\mathcal{E}$ given on the penultimate line of p2 is wrong: it implies that $\mathcal{E}$ is a number, but it is actually a function of the many- boson position vector $\mathbf{x}$. Please fix this.

  • validity: high
  • significance: good
  • originality: ok
  • clarity: good
  • formatting: good
  • grammar: good

Author:  Mathias Gartner  on 2022-06-03  [id 2554]

(in reply to Report 2 on 2022-04-06)

We thank the referee for reading the manuscript and his/her report. In the following we address the questions and comments raised:

  1. We understand that these could indeed be a bit confusing. The reason for writing the potential in this form is the usual convention in cold atomic systems where the optical lattice potential is generated by counter propagating laser beams with wave vector $k_L$, which results in the lattice of the given form. Basically, we follow the literature (eg. Bloch et al. 10.1103/RevModPhys.80.885, Greiner et al. 10.1038/415039a) and books (eg. Lewenstein et al. “Ultracold Atoms in Optical Lattices”). We therefore want to keep this notation. However, we understand the possible confusion this may lead to, and have added a sentence explaining its origin in the revised version.

  2. The referee raises valid questions about obtaining information on the excitations by relying only on Monte Carlo noise. If we would sample from the exact trial wavefunction, the local energy would be the exact ground state energy and have no variance. Then the right hand side of the differential equation (6) for the variational parameters would vanish (while $S_{KK’}$ would still fluctuate). Since $S_{KK’}$ is invertible, all $\dot\alpha_K$ vanish, hence indeed all $\alpha_K$ stay constant exactly (in contrast to approximate trial functions where $\alpha_K$ keeps on fluctuating slightly during the time evolution). When we sample the density from this exact ground state, the stochastic noise of the density will be purely due to the Metropolis sampling, and particularly will be uncorrelated in time. We have tried this for non-interacting particles where the exact wave function is known, and indeed we do not see the single particle dispersion in the power spectrum of the density fluctuation. So the referee is completely right in that when working with the exact ground state one cannot use the MC noise to get the excitation spectrum. We note however, that for most many-body problems the exact ground state is beyond reach. We have improved the trial wave function by using more parameters $\alpha_K$ (and still using the Jastrow ansatz) and find that the reduced variance of the local energy indeed lowers the signal-to-noise ratio for the excitation peaks in the density fluctuation power spectrum, however, the peak positions do not change. If on the other hand we used more samples, the whole power spectrum is scaled down but the signal to noise ratio is not affected by sample size. We have added a comprehensive discussion in the manuscript in the second paragraph of the new section 3.3.

  3. To give a more clear statement about the choice of the trial wavefunction we write the Jastrow ansatz as exponentials of correlation functions - which is still a completely general ansatz. Furthermore we extend the paragraph where we link the parametrized wavefunction to this ansatz.

  4. Thank you for pointing this out. We fixed it.

---

## Round 1 · Referee Report · Anonymous (Referee 3) · 2022-4-7

Strengths

  1. First calculation (to my knowledge) of the dynamical structure factor using t-vmc

  2. Ability to access the regime of non-linear perturbation

Weaknesses

  1. The strategy to compute S(q,omega) with "noise only" is not too well justified in the paper

Report

This is a very interesting work studying the time evolution of interacting bosons . The work is novel, especially on the technical level, and I strongly recommend publishing it once the points below have been addressed.

  1. In order to understand the accuracy of the method, it might be useful to first report the accuracy on the ground state results in the absence of the external field. What is the accuracy compared to the Bethe ansatz solution of Lieb Liniger model?

  2. While the form of the external potential is explicitly given, the authors do not specify how the delta interaction is implemented/ regularized in practice. Do the authors use an effective two-body potential or something similar?

  3. The discussion on obtaining the S(q,omega) directly from the monte carlo noise is very interesting and suggestive. However, I find that the lack of mathematical analysis and justification somehow problematic, indeed the main issue I see is that the noise among different parameters is going to be correlated among different parameters, and the propagation of the noise in the ODE is also highly non-linear and non-trivial. I am not sure this going to yield an unbiased estimator of S(q,omega). I would suggest to either rephrase the discussion to reflect this formal uncertainty or remove entirely this part.

  4. (Optional) Would it be possible to compare the dynamics more quantitatively to what obtained with the Bose Hubbard model ? The authors report about qualitative similarities, but I believe it would be nice to see also a direct comparison with exact diagonalization results for the dynamics of the Bose Hubbard, also to understand how accurate is the t-VMC in this specific case.

Other points

a. How is the matrix S regularized / is it regularized at all? b. Since there is no specific length limitation here (I believe) it would be useful to add a few more details on the sampling procedure used / ODE solver etc , also in the interest of reproducibility

  • validity: high
  • significance: high
  • originality: high
  • clarity: good
  • formatting: good
  • grammar: excellent

Author:  Mathias Gartner  on 2022-06-03  [id 2555]

(in reply to Report 3 on 2022-04-07)
Category:
answer to question

We thank the referee for reading the manuscript and his/her report. In the following we address the questions and comments raised:

  1. The variational ground state energy of the Lieb Liniger model compares very well with Bethe ansatz calculations. We benchmarked our results to the energy fit given in the publication by Lang et al. (10.21468/SciPostPhys.3.1.003), equation (10), and obtained a relative energy error for our variational wavefunctions of less than 0.4 percent for the interaction strengths that are relevant in this work. We include this check in the revised manuscript.

  2. We briefly mentioned in section 2.1 that we incorporate the contact interaction as a boundary condition on the wavefunction. To clarify this, we expand this paragraph by giving the required boundary condition and mention that the variational parameters are restricted in such a way that this condition is always fulfilled. In particular we use $\frac{1}{\Phi} \frac{\partial}{\partial x_i}\Phi = \frac{1}{4} k_L g$.

  3. Our results show that the time-dependent density correlations due to noise provide the same location for peaks associated with excitation as does $S(k,\omega)$. This may not be surprising, because the random fluctuations on top of the variational ground state are just a superposition of many-body exciations. But we do not claim that we get the whole $S(k,\omega)$ only from noisy ground state propagation in tVMC: to calculate $\operatorname{Im} \left[ \delta\tilde\rho(k, \omega) / \delta \tilde V_p(k, \omega) \right]$, we would need $\delta \tilde V_p(k,\omega)$ which we cannot quantify if we use Monte Carlo noise. For this reason we show the power spectrum $|\delta\tilde\rho(k, \omega)|$ of the density fluctuations, which has excitation peaks where $S(k,\omega)$ has peaks. Also in reply to a question of referee 2 about the noise method, we discuss this much more comprehensively in the new section 3.3. In particular we state explicitly that for calculating $S(k,\omega)$ with all the correct spectral weights rather than just peak locations, we use the linear response method.

  4. The dynamic structure function $S(k,\omega)$ for the Bose Hubbard model can be found in Roux et al (previously [35], now [39]). We use the same $U/J$ ratio to allow the comparison between our variational result on a continuous optical lattice and the presumably exact BHM result in [39] obtained with exact diagonalization, albeit with smaller $N=16$ than our $N=20$. As discussed in the manuscript, the overall shape of $S(k,\omega)$ and resulting excitation energies in [39] are essentially the same as our result, but the former is more structured, with multiple close peaks for $k$ around the edge of the Brioullin zone, compared to our single peak in $S(k,\omega)$. This may be due to the limitation of our Jastrow ansatz, but also due to different particle numbers. Also note that [39] accounts for only the lowest band.

Other points: a. We precondition and regularize the matrix $S$ by scaling the entries according to ${S_{KK'}' = S_{KK'}/\sqrt{S_{KK}S_{K'K'}}}$ and adding a small value $\varepsilon = 10^{-4}$ to the diagonal. b. We use QR decomposition for solving the equation system (6) and a fourth order Runge Kutta scheme to propagate the ODE in time. All the expectation values are estimated by means of Monte Carlo integration where we use the Metropolis-Hastings algorithm for generating uncorrelated samples. To provide this information in the revised submission, we have added a new section 2.2 where we give further details on the numerical methods.

---

## Round 2 · Referee Report · Anonymous (Referee 2) · 2022-6-5

Report

I am satisfied with the authors' response to my report and the other reports and recommend publication. The paper meets expectation number 3: it "opens a new pathway in an existing or a new research direction, with clear potential for multipronged follow-up work". It also meets all of the general acceptance criteria.

---

## Round 2 · Referee Report · Anonymous (Referee 3) · 2022-6-6

Report

I thank the authors for clarifying the points I raised in my original reports and for improving the quality of the manuscript. I believe the paper is now ready to be published.

---

## Round 2 · Referee Report · Anonymous (Referee 1) · 2022-6-7

Report

I think the the authors addressed all previous concerns and the manuscript is now suitable for publication in SciPost Physics.

---

## Round 2 · Author Response

Dear Editor,

Please find our resubmission of the manuscript which includes the suggestions raised by the referees. We hope that they are adequate and suitable to warrant publication in the SciPost Physics journal.
The answers to the referees' questions can be found as comments to the reports.

Yours sincerely,
Mathias Gartner, Ferran Mazzanti and Robert E. Zillich

---

## Round 2 · List of Changes

Taking into account the referees’ suggestions, we made the following changes to the manuscript:
- change the title to “Time-dependent variational Monte Carlo study of the dynamic response of bosons in an optical lattice”
- dividing the manuscript in sections
- corrected the expression for the local energy $E=H \Psi / \Psi$ [Report 2.4]
- in abstract distinguish between the power spectrum of density fluctuations and the dynamic structure factor [Report 1.1]
- we now refer to $|\delta \tilde \rho (k, \omega)|$ as the square root of the power spectrum of the density fluctuations
- introduction to our results at the end of the new section “Introduction”
- add explanation why we use the form $sin^2(k_L x)$ as optical lattice potential [Report 2.1]
- add information about the boundary condition on the wavefunction due to the contact interaction [Report 3.2]
- add color legend in Fig. 2 [Report 1.2]
- add more detailed discussion about the comparison with [39] in section 3.1 [Report 1.2]
- in section 3.3 we significantly extend the discussion of the noise in equation (6) [Report 2.2, Report 3.3]
- write the Jastrow ansatz as products of the exponential of correlation functions and extend the description of our variational ansatz for the wavefunction [Report 2.3]
- add new section 2.2 where technical details of the numerical simulations are stated [Report 3]
- we compared the ground state energy of our variational wavefunction for a Lieb Liniger model to Bethe ansatz calculations and state the result in section 3 [Report 3.1]
- correct an error in the $\omega$-axis values in figure 3
- add new references [24], [30], [37] and [38]
- generally we tried to improve readability by changing wording throughout the text
- adding label (a), (b), and (c) for the panels in Fig.3

---

## Editorial Decision

published